# Prevention of child wasting: Results of a Child Health & Nutrition Research Initiative (CHNRI) prioritisation exercise

**Severine Frison**[1,2]*, **Chloe Angood**[1], **Tanya Khara**[1], **Paluku Bahwere**[3], **Robert E. Black**[4], **André Briend**[5], **Nicki Connell**[6], **Bridget Fenn**[1], **Sheila Isanaka**[7,8,9], **Philip James**[10], **Marko Kerac**[10], **Amy Mayberry**[11], **Mark Myatt**[12], **Carmel Dolan**[1], on behalf of the wasting prevention Working Group Collaborators[¶]

1 Emergency Nutrition Network, Oxford, United Kingdom, 2 Department of Infectious Disease and Epidemiology, London School of Hygiene and Tropical Medicine, London, United Kingdom, 3 Valid International, Oxford, United Kingdom, 4 Institute for International Programs, Bloomsbury School of Public Health, Johns Hopkins University, Baltimore, United States of America, 5 Center for Child Health Research, Tampere University, Tampere, Finland, 6 Eleanor Crook Foundation, Washington DC, United States of America, 7 Department of Nutrition, Harvard School of Public Health, Boston, United States of America, 8 Department of Global Health and Population, Harvard School of Public Health, Boston, United States of America, 9 Department of Research, Epicentre, Paris, France, 10 Department of Population Health, London School of Hygiene and Tropical Medicine, London, United Kingdom, 11 No Wasted Lives and Action Against Hunger UK, London, United Kingdom, 12 Brixton Health, Llwyngwril Gwynedd, United Kingdom

¶ The list of the wasting prevention Working Group Collaborators can be found in the Acknowledgments
* severine.frison@ennonline.net,

## Abstract

### Background

An estimated 49.5 million children under five years of age are wasted. There is a lack of robust studies on effective interventions to prevent wasting. The aim of this study was to identify and prioritise the main outstanding research questions in relation to wasting prevention to inform future research agendas.

### Method

A research prioritisation exercise was conducted following the Child Health and Nutrition Research Initiative method. Identified research gaps were compiled from multiple sources, categorised into themes and streamlined into forty research questions by an expert group. A survey was then widely circulated to assess research questions according to four criteria. An overall research priority score was calculated to rank questions.

### Findings

The prioritised questions have a strong focus on interventions. The importance of the early stages of life in determining later experiences of wasting was highlighted. Other important themes included the identification of at-risk infants and young children early in the progression of wasting and the roles of existing interventions and the health system in prevention.

---

**Data Availability Statement:** All relevant data are available at https://doi.org/10.5281/zenodo.3631220.

**Funding:** This paper was produced through the Maximising the Quality of Scaling Up Nutrition Plus (MQSUN+) project supported by UK aid and the UK Government; however, the views expressed do not necessarily reflect the UK Government's official views or policies. This paper was also funded by PATH through contract DFI.2118-01629955-CRT by UK Aid from the UK government. The funders had no role in study design, in the data collection, the analysis, interpretation of data, and in the decision to submit the paper for publication.

**Competing interests:** The authors have declared that no competing interests exist.

## Discussion

These results indicate consensus to support more research on the pathways to wasting encompassing the in-utero environment, on the early period of infancy and on the process of wasting and its early identification. They also reinforce how little is known about impactful interventions for the prevention of wasting.

## Conclusion

This exercise provides a five-year investment case for research that could most effectively improve on-the-ground programmes to prevent child wasting and inform supportive policy change.

## Introduction

There are an estimated 49.5 million wasted children under five years of age [1]. The decline in the global prevalence of wasting has been slow, from 7.9% in 2012 to 7.3% in 2018; just 37 (19%) out of 194 countries are on track to achieve the World Health Assembly (WHA) 2025 target of maintaining prevalence of wasting below 5.0% [2]. World hunger appears to be on the rise after a prolonged decline [3]. Although wasting is commonly considered an acute condition due to its relatively rapid onset and resolution compared to other manifestations of undernutrition such as stunting, the contributing factors and effects can be long term [4]. A recent analysis of the WHA targets highlighted the lack of robust studies on effective interventions to prevent wasting and a strong tendency in the global nutrition community to focus on stunting prevention and wasting treatment rather than wasting prevention [5].

Failure to address wasting has significant consequences both for individual children and communities. Of all forms of malnutrition, it has the highest short-term case fatality rate [6, 7]. This is especially true in its most severe form or when combined with stunting. Severely wasted children aged 6 to 59 months are 9 to 12 times more likely to die than their healthy counterparts [8, 9]. Wasting is also a problem earlier in life [10], although similar estimates of its contribution to mortality are not available for infants under six months. Also important are long-term sequelae including developmental/cognitive deficits [11] and increased risk of non-communicable diseases (NCDs) in later life [12–14]. Immune dysfunction is both a cause and a consequence of malnutrition, contributing directly to the mortality and morbidity associated with wasting [15, 16]. There is also emerging evidence that wasting is a 'harbinger of stunting', whereby linear growth is impaired by episodes of wasting [17]. Thus, lack of progress in tackling wasting may also affect progress towards the WHA stunting target [14, 17–19]. Although remarkable improvements have been made over the last two decades in treating severe wasting, an understanding of the fundamental risk factors, mechanisms and pathophysiological changes contributing to the condition's development remains limited. This knowledge gap critically hampers the ability to prevent wasting in the first place [20].

Recognising the limitations in understanding, as outlined above, and appreciating the limited time and resources available to tackle a globally important public health condition, the aim of this study was to identify and prioritise the main outstanding research questions/gaps in relation to wasting prevention.

## Methods

This research prioritisation (RP) exercise followed the Child Health and Nutrition Research Initiative (CHNRI) method, described in detail elsewhere [21, 22], developed to assist stakeholders in prioritising health research investments. The method involves identifying and listing a large number of possible research options within a well-defined context, based around a "4D" framework, by which research questions are grouped into four themes: Description, Delivery, Development and Discovery. In this case, 'Description' includes research to assess the burden of wasting and its determinants; 'Delivery' encompasses research to prevent wasting, using already available interventions; 'Development' describes research to improve existing interventions to better prevent wasting; and 'Discovery' includes research that may lead to innovations/ completely new interventions. Using these four themes ensures consideration of a wide breadth of possible research options. The method then allows for a systematic, transparent and structured means for experts to score these possible research options against predefined and relevant criteria. The result is a prioritised list of research questions that can be used by international agencies, donors, national governments and policy-makers to stimulate dialogue and inform investments in research in the subject area [21].

Guided by this method, the context and scope of this RP exercise was outlined by the core research team and, drawing from recent reviews on the aetiology of wasting [4] and the current state of evidence and thinking on wasting prevention [23] as well as previous related CHNRI exercises [24–26], an initial list of 94 research questions was developed. An expert group (EG) of leading specialists in nutrition, infant and child growth and epidemiology collectively refined and reduced the list to 40 key research questions, organised by the '4Ds'. The group agreed that the focus of the RP exercise would be on research that could provide results within a five-year period, for infants and young children 0 to 59 months of age, living in low and middle income countries; would consider wasting as well as other forms of acute malnutrition (e.g. bilateral oedema, low mid-upper arm circumference (MUAC) and low weight-for-age); and take a broad view of prevention, i.e. considering preventing any wasting as well as any worsening of its severity (S1 File). The group also selected four criteria from those recommended by the CHNRI process, against which the questions should be judged (Table 1), considering the topic of wasting prevention. A target list of participant profiles for the subsequent prioritisation survey was also drawn up to reflect a broad spread of geography (global, regional, country, sub-national), types of organisations and areas of expertise.

A survey was developed using the online tool 'SurveyMonkey' (www.surveymonkey.co.uk) with question order randomised by the four "Ds" to ensure a similar response rate for each section of questions. Following a short pilot, corrections and adjustments were made, after which the final survey was made available from November 2018 to February 2019. The survey link was circulated via the EG, the Incidence of Acute Malnutrition group, the Management of At-risk Mothers and Infants (MAMI) group, the Wasting and Stunting (WaSt) group (n = 828),

**Table 1. Selected criteria.**

| | |
|---|---|
| Answerability | How answerable would this research question be? (e.g. is it feasible to answer within the given context and timeframe? Is it ethical?) |
| Efficacy | How likely is it that this research would lead to efficacious interventions/approaches/policies? (e.g. is it likely to produce the desired outcome in ideal conditions?) |
| Deliverability | How likely is it that this research would lead to deliverable impactful interventions/approaches/ policies? (e.g. will the intervention/ approach/ policy be affordable, cost-effective, deliverable at scale and achieve required coverage? Is the research generalizable?) |
| Fills a gap | Will this research question fill a key gap in knowledge that is required to prevent wasting? |

circulation lists from No Wasted Lives [27], the Department for International Development (DFID), Global Nutrition Report–Independent Expert Group, Health Systems groups, International Lipid-Based Nutrient Supplements (iLNS) project lists, Maximising the Quality of Scaling Up Nutrition⁺ (MQSUN⁺ subscribers' list), Leveraging Agriculture for Nutrition in South Asia (LANSA) consortium, cluster coordinator lists (health, WASH, food security), the United Nation's Children fund (UNICEF) publications sharing list; and the ENN website [28] and social media accounts.

Survey participants were asked to consider future interventions resulting from the stated research questions, and judge how each question might meet each of the four criteria. For each question, participants were required to judge if each of the criteria were met by indicating "Yes" (which was then allocated 1 point), "Undecided" (0.5 points), "No" (0 points), or "Insufficiently informed" (no input). A research priority score (RPS) of 0–100% was calculated for each criterion for each research question; from this, an overall RPS for each question was computed (the mean of the RPS for each criterion). The level of agreement between respondents' answers was assessed through the average expert agreement (AEA), a proportion of scorers who gave the most common score (mode) for a question divided by the total number of scorers who scored that question, as follows:

$$AEA = \frac{1}{4} X \sum_{q=1}^{4} \frac{N(\text{number of scorers with most frequent})}{N(\text{number of scorers who provided any answer})} x100$$

where q is a question that experts are being asked to evaluate and 4 is the number of answers that can be given.

The AEA is unaffected by responses of 'undecided' and is also unaffected by the varying number of scorers per criterion and differences in scorer composition for the different criteria. In AEA computation, all four possible responses ("Yes", "No", "Undecided", or "Insufficiently informed" (no input)) are treated as valid. Therefore, if a substantial proportion of the experts respond as "insufficiently informed", the AEA will reflect this and reduce the level of overall agreement, rather than increase it.

## Ethics

As is standard for CHNRI exercises [24, 25], this project does not require formal ethical committee review. The work does not involve medical research on human subjects, no personal or sensitive data was used and it involved professional participants rather than patients. All participants were invited to participate in the CHNRI exercise through a variety of platforms and by responding to the invitation, they acknowledged their voluntary participation in the exercise and no special informed consent was required. Participants who completed the survey were asked whether they were happy to be part of wasting prevention Working Group Collaborators list; those who answered "yes" are named in the acknowledgment. Furthermore, all input received from participants was encoded and no identifiable information was linked to the participant's submissions. All data is anonymous and participants were informed that data would be used for publication.

## Results

### Characteristics of the respondents

In total, 146 individuals participated in the survey, with an average completion rate of 83% (ranging from 5% to 100% of questions, with 34 participants (23%) completing the entire survey, 113 (77%) completing over 80% and 122 (84%) completing over 60%). At least 108

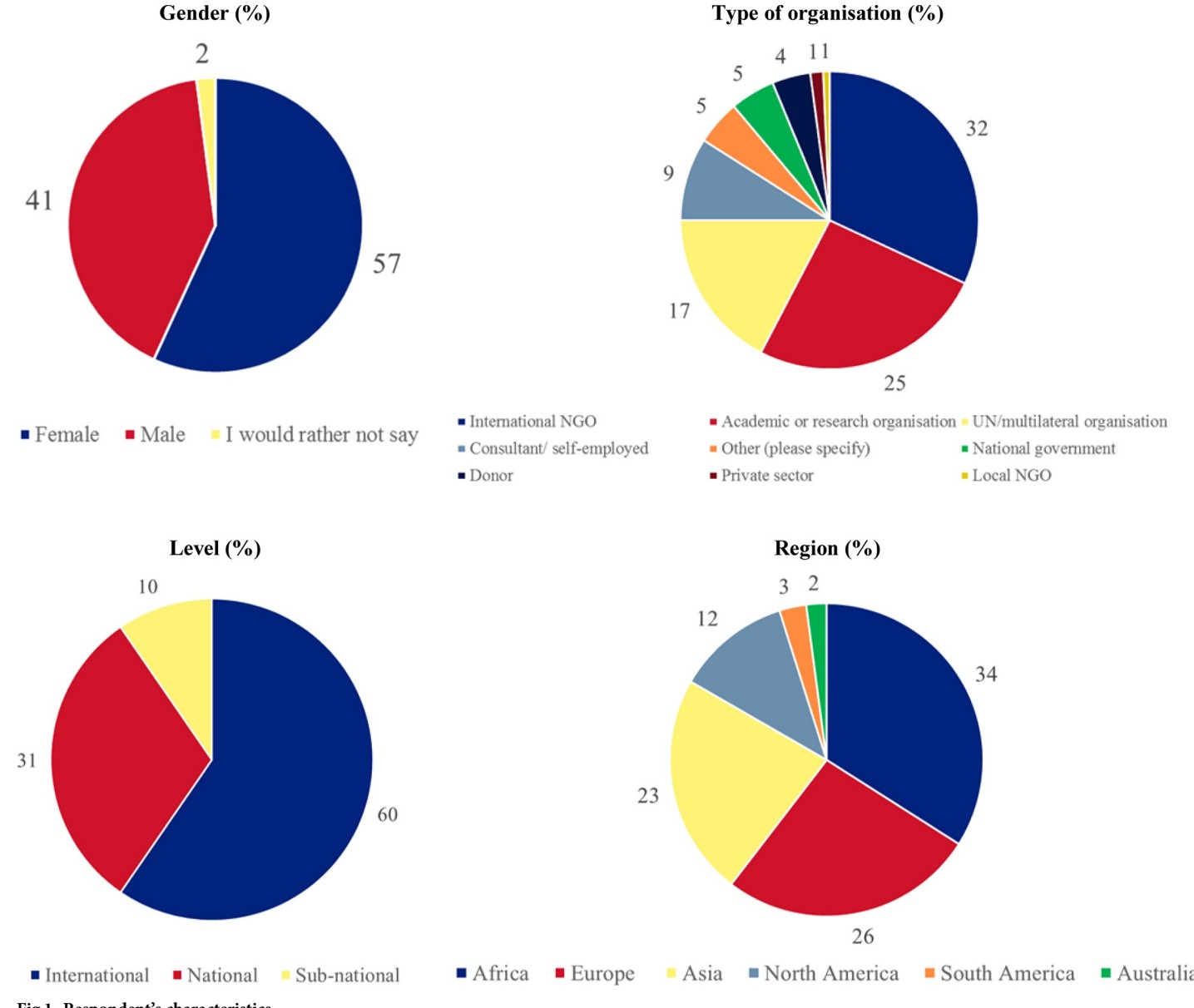

**Fig 1. Respondent's characteristics.**

participants responded to each question (ranging from 108 to 131 (S1 Table)) and over half (n = 83, 57%) of the participants were female. Respondents worked at a range of levels including international (n = 88, 60%), national (n = 45, 31%) and sub-national (n = 15, 10%). Most respondents worked in programme implementation with a third working for non-governmental organisations (NGOs) (n = 47, 32%), a fifth for United Nations (UN) agencies (n = 25, 17%) and a small proportion for national governments (n = 7, 5%). Research organisations represented a quarter of respondents (n = 37, 25%). There was representation from all regions of the world, but in particular Africa (n = 50, 34%), Europe (n = 38, 26%) and Asia (n = 34, 23%) (Fig 1).

**Table 2. Top ten research questions according to the overall research priority score (RPS).**

| Rank | Research question | Group | A | E | D | F | RPS | AEA | N |
|---|---|---|---|---|---|---|---|---|---|
| 1 | What is the impact of interventions for managing at-risk mothers and infants less than 6 months of age in preventing wasting/acute malnutrition in the >6months old? | Description | 97.5 | 96.7 | 93.9 | 95.9 | 96.0 | 93.1 | 122 |
| 2 | What is the impact of growth failure during the first 6 months of life on experience of wasting/acute malnutrition after 6 months of age? | Description | 96.3 | 88.0 | 88.9 | 89.3 | 90.6 | 86.4 | 117 |
| 3 | How can existing interventions (e.g. growth monitoring, integrated management of childhood illness (IMCI)) better detect and support children (0–59 months) who are failing to thrive/faltering (i.e. those at-risk, not just those already below a z-score threshold)? | Development | 91.4 | 90.1 | 89.8 | 87.8 | 89.7 | 85.6 | 131 |
| 4 | What are effective and cost-effective approaches to target the highest risk infants and children 0–59 months (e.g. children with concurrent wasting/acute malnutrition and stunting, children <24 months, etc.) for interventions (food or non-food) to prevent wasting/acute malnutrition? | Delivery | 93.3 | 89.4 | 85.4 | 88.4 | 89.1 | 84.7 | 129 |
| 5 | What measures (anthropometric or non-anthropometric) or combinations of measures best identify individual infants and children (0–59 months) by age/sex at most risk of death/other adverse outcomes associated with wasting/acute malnutrition? | Description | 91.9 | 89.2 | 87.4 | 85.5 | 88.5 | 82.8 | 128 |
| 6 | What is the role of pre-pregnancy maternal factors (age, health status, nutritional deficits, psychological factors etc.) in determining risk of being born with a low birth weight, low weight-for-length, low mid-upper-arm circumference, premature or small for gestational age? | Description | 96.7 | 87.6 | 86.4 | 83.2 | 88.5 | 83.2 | 122 |
| 7 | What measures (anthropometric or non-anthropometric), or combinations of measures, best identify individual infants and children (6–59 months) by age/sex at most risk of wasting/acute malnutrition? | Description | 92.3 | 88.2 | 89.0 | 81.3 | 87.7 | 82.0 | 128 |
| 8 | What are effective and cost-effective approaches to integrating wasting/acute malnutrition prevention efforts into health systems (i.e. human resource capacity, financing, supplies and supply chain, etc)? | Delivery | 86.0 | 89.9 | 84.3 | 89.5 | 87.4 | 81.0 | 128 |
| 9 | What impact can effective wasting/acute malnutrition prevention interventions/approaches have on levels of stunting (and concurrent wasting and stunting) and vice versa? | Description | 89.5 | 85.5 | 85.0 | 87.2 | 86.8 | 79.5 | 125 |
| 10 | How does being born prematurely and/or with foetal growth restriction impact on wasting/acute malnutrition at birth and throughout the first 5 years of life, by sex? | Description | 93.0 | 81.6 | 83.8 | 86.5 | 86.2 | 80.2 | 122 |

A = answerability; E = efficacy; D = deliverability; F = fills a gap; RPS = overall research priority score; AEA = average expert agreement; N = number of respondents

## Research priority questions

The top ten research priority questions according to the overall score are presented in Table 2. The AEA in the top ten questions was high (79.5% to 93.1%) and varied from 65.6% to 93.1% overall (see Table 2 and S2 Table), indicating a high level of agreement among respondents. The number of respondents (ranging from 117 to 131 for the top ten questions) was slightly larger the higher the AEA (Table 2). A theme common to the highest priority questions was exploring the importance of the early stages of life (pre-pregnancy, *in utero* and in the first six months of life) in determining later experiences of wasting (from six months of age upwards) and effective interventions during these critical periods. Specifically, the top two ranked questions focus on infants under six months of age and the extent to which this critical period and interventions targeted at them and their mothers have a bearing on subsequent wasting. Questions ranked three and four relate to the identification of effective approaches, including the potential of existing interventions to detect and support at-risk infants and children to prevent wasting. The remaining priority questions concern how to best identify at-risk infants and young children (Question ranked 5, Question ranked 7); the role of pre-pregnancy and foetal factors in wasting of infants and children (Question ranked 6 Question ranked 10); the role of existing interventions and the health system in preventing wasting (Question ranked 8); and exploring dual impacts on both wasting and stunting that prevention approaches may achieve (Question ranked 9). The ranking of all 40 questions is provided in S2 Table.

The questions prioritised have a strong focus on interventions. Two of the 'Description' questions that ranked in the top ten (ranked 1 and 9) are on the impact of existing

**Table 3. Research questions by the '4Ds': Description, Delivery, Development and Discovery.**

| Question groups (4Ds) | Questions N | Questions in top ten N | Proportion in top 10 (%) |
|---|---|---|---|
| Description | 18 | 7 | 39 |
| Delivery | 7 | 2 | 29 |
| Development | 4 | 1 | 25 |
| Discovery | 11 | 0 | 0 |

interventions, while two others (ranked 5 and 7) relate to ways to target individuals or the timing of interventions. Along with the three 'Delivery' and 'Development' questions, over half of the top ten questions related to current interventions that need to be evaluated or possibly improved.

Seven of the top ten priority questions fell under the 'Description' theme (research to assess the burden of wasting and its determinants). Only two questions in the top ten came under 'Delivery' (research to assist in the optimising of the nutrition status of the population through existing delivery models) and one under 'Development' (research to improve interventions that already exist). Questions from the 'Delivery' and 'Development' groups that ranked in the top ten related to the integration of wasting prevention efforts into existing routine programmes (ranked third overall) and health systems (ranked eighth overall) and the need to target interventions to those most at risk (ranked fourth overall). There was also interest in examining models of community engagement, the use of food-based compared to product-based approaches and the benefit of nutrition-sensitive versus nutrition-specific approaches.

No 'Discovery' questions (research leading to innovation/ new health interventions) were ranked within the top 10 (Table 3) and the AEA was lowest of all in this group of questions (mean and median of 70% and 69% respectively compared to a AEA mean and median of 79%). The first Discovery question, "What programmatic or project-based innovations (across all sectors / multi-sectoral) have led to prevention of wasting/acute malnutrition in a given context?" is ranked 20 out of 40 and the second Discovery question, ranked 28, relates to the impact of multiple uses and management of water resources on wasting (see the ranking of questions by Ds in S2 Table).

Many questions ranked last had a physiological focus which could be due to the fact they were too clinical for respondent's interests. Questions related to context (for example, questions around governance, policies, institutions, environment and systems strengthening) scored lower than questions focusing on wasted individuals and their mothers.

## Discussion

This RP exercise aimed to fill a critical gap by generating consensus around research questions that could enable the international and national communities to move forward with effective wasting prevention strategies. Results point to the need for research to establish 1) whether interventions targeting pre-birth and initial weight loss in early infancy and childhood can reduce the subsequent risk of wasting and 2) interventions that can better identify and target those most at risk of death. The results also highlight the need to determine how existing service delivery systems can be better harnessed by integrating wasting prevention approaches within them. These findings accord with the "first thousand days" initiatives [29] and the growing global evidence on the need to prevent and catch wasting early to both maximise survival and prevent deleterious effects impairing a child's ability to thrive [16]. It is also consistent with recent evidence on the need to link nutrition with neonatal care [30, 31], the high

burden of wasting in early infancy [32], the strong association between foetal growth restriction and subsequent wasting on persistent wasting [33] and the risk that an episode of wasting can lead to subsequent episodes [34, 35]. The geographical and organisational spread of respondents [24–26] implies that the results represent broad expert consensus of global priority areas. The high level of agreement in the ranking of the research questions, in the top ten especially, gives confidence in the identified priorities.

A high proportion of prioritised questions fell in the 'Description' group, reflecting the fundamental gap in the basic understanding of wasting, its causes and how to identify those at most risk [4]. The results therefore echo the earlier conclusion that progress has been made in the treatment of wasting (and therefore the mortality associated with wasting), but current understanding of its aetiology remains limited [4, 20]. A recent review of countries where wasting levels have remained high despite numerous interventions highlights the limited understanding of the pathways to wasting and, therefore, the optimal points at which to intervene and prevent it [33]. A high number of priority questions also focussed on the improvement and/or measurement of the impact of current interventions. This gap in knowledge on what works to prevent wasting was also highlighted in a recent report that concluded that many interventions may have an effect, but that this effect is not currently measured [23].

Although some of the discovery questions were very topical (e.g. question 36 on the physiological factors that could explain the multiplicative effect of wasting and stunting on mortality and question 30 on the effect of the microbiome and environmental enteric dysfunction on wasting prevention), none of them were ranked highly. This may be due to the more controversial nature of innovative questions. It also demonstrates that the participants assigned more value to practical research questions with more immediate operational effect, rather than questions that could take longer to answer and operationalise. The specified context of a five-year timeframe may have influenced this. The lower AEA in the 'Discovery' questions also suggests more variation in the prioritisation of these questions, possibly due to their more specialised nature.

The last ten prioritised research questions having a more physiological focus could reflect the characteristics and background of the respondents, as the majority of respondents were on the more technical operations and implementation side of nutrition programmes. Questions focused on context and political economy (governance, policies, institutions, environment, and health systems strengthening) also scored lower than research questions on wasted children and their mothers, which may reflect the CHNRI process and the people involved, or it could be because these are more challenging to answer and require more innovative, context specific and community-based approaches to address.

This study had several strengths. It used a validated approach usually associated with good reproducibility [36]. It had good geographical coverage and most respondents were on the frontline of the fight against wasting. It was a systematic, transparent and structured process that provides five-year investment priorities for research that could effectively inform policy change and related on–the-ground practice.

There are also, however, limitations. While the CHRNI process is widely used, useful and practical, it also risks simplifying complex problems and only represents those who respond. It is unknown how representative these are of a wider population of stakeholders. In particular, selection biases can occur.

Firstly, bias may have arisen from the initial selection of questions. The list of possible research questions could never be exhaustive, due to the extremely broad area of research covered. Nevertheless, care was taken to ensure the broadest spectrum of questions possible by gathering them from a wide range of sources, including recent related review [4, 23] and other RP exercises [24, 37]. For practical reasons this list had to be consolidated further, so as not to

deter respondents by requiring them to complete a very lengthy survey. While the authors made effort to ensure a breadth of expertise in the group that advised on the re-wording and honing down of the original 94 questions to 40, an emphasis on more medicalised/interventionist approaches to addressing wasting in the questions selected is noted, with less emphasis on the wider social, economic and environmental determinants, or political economy of nutrition. A broader group of experts such as economists and social scientists may have led to a broader spectrum of questions included.

Bias may also have been introduced in the self-selection of respondents. Due to the online nature of the survey and invitations by email using existing mailing lists, only those with reliable internet access engaged in global platforms and networks were likely to participate, resulting in further selection bias. This may be reflected in the higher representation of academics and NGO and UN staff among respondents rather than government representatives and those working at sub-national levels. The high agreement rate in the results may also demonstrate a narrow base of respondents. However, the survey captured a broad range of experts working in nutrition programming and policy and no critical missing groups were identified. Furthermore, the very nature of a RP exercise gathers the opinions of individuals who are engaged and interested in the topic area, which is one of the strengths of the process. There is also an underrepresentation of respondents from Latin American countries, however, this probably reflects the fact that wasting is not commonly acknowledged as a substantial problem in that region.

## Conclusion

Wasting is a critical public health problem. While some progress has been made in the reduction of stunting, the prevalence and burden of wasting has barely changed from 2011 levels.

This RP exercise highlights knowledge gaps that have limited the ability of international and national actors to prevent wasting and achieve related global nutrition targets. These results indicate consensus to support more research on the pathways to wasting encompassing the *in-utero* environment, the early period of infancy and a focus on the process and identification of wasting (rather than the state of being wasted). They also reinforce how little is known about impactful interventions for the prevention of wasting and underline the need for research and evaluation to move beyond a focus on single forms of undernutrition, ensuring that there is equal attention given to wasting, as to other forms of malnutrition, wherever it is present. This CHNRI provides a five-year investment case for research that could most effectively inform policy change and which is closely related on-the-ground practice. Donors, international and national organisations, governments and research institutions can use these results to inform more coherent research investments in this critically important area.

## Supporting information

**S1 File. Wasting Prevention Research Prioritisation–Scope, Definitions and Context.**
(DOCX)

**S1 Table. Research questions ranking according to the overall research priority score (RPS).**
(DOCX)

**S2 Table. Research questions ranking by Ds according to the overall research priority score (RPS).**
(DOCX)

## Acknowledgments

We would like to thank the additional members of the Wasting Prevention CHNRI Expert Group who contributed to this work, namely Zulfiqar Bhutta, Saskia DePee, Andrew Hall, Jeanette Bailey, Emily Smith and Helen Young. We would also like to thank the wasting prevention group collaborators namely Caroline Abla, Jogie Abucejo Agbogan, Jasinta Achen, Tammam Ahmed, Muhammad Mazhar Alam, Bernice Amadotor, Satinder Aneja, Amador Gómez Arriba, Shewangizaw Ashenafi, Sufia Askari, Cecile Basquin, Elodie Becquey, Katie Beck, Jay Berkley, Adane Beyene, Nita Bhandari, Rita Bhatia, Bindi Borg, Barbara Bobba, Anne Bush, Main M. Chowdhury, Bernadette Cichon, Jean Marius D'Alexandris, Abner Elkan Daniel, Hedwig Deconinck, Nicky Dent, Samson Desie, Agnès Dhur, Alison Donnelly, Wisdom Dube, Tuoch Duoth Kang, Tatyana El-Kour, Colleen Emary, Montse Escruela, Fernando Fernandez, Valerie Flaherman, Suzanne Fuhrman, Maureen Gallagher, Masunne Prosper Galseku, Wondayferam Gemeda Guluma, Grana Pu Selvi Gnanaraj, Ted Greiner, Carlos Grijalva Eternod, Fakhar Gulzar, Lieven Huybregts Phuong Huynh, Alessandro Iellamo, Jo Jacobsen Collins John, Ateca Kama, Seema Kapoor, Shivangi Kaushik, Emily Keane, Anika Krstic, Priti Kumari, Natasha Lelijveld, Crepin Louhoungou, Rolf Luyendijk, Hussain Bux Mallah, Mark Manary, Karim Manji, Esther Matama, Anuradha Manthripragada, Marie McGrath, Abdulahi Or Mohamed, Warsame Said Mohamed, Alex Mokori, Grainne Mairead Moloney, Assumpta Mukabutera, Martha Nakakande, Tina Kaonga Nyirenda, Sarah O' Flynn, Gloria Obeng-Amoako, Hellen Okochil, Orla O'Neill, Susan Onyango, Danka Pantchova, Kagayo Paul, Kevin Phelan, Ramesh Poluru, Babikene Rasi, Kate Reinsma, Julie Rop, Khrist Roy, Alexandra Rutishauser-Perera, Cécile Salpéteur, Priscilla Scariah, Helene Schwartz, Andrew Seal, Kebeda Shitaye, Zay Ya Soe, Ruth Situma, Tor Strand, Alison Talbert, Casie Tesfai, Indi Trehan, Sophiya Uprety, Saskia van der Kam, Anne Walsh, Henri Wamani, Patrick Webb, Zita Weise Prinzo, Sophie Whitney, Caroline Wilkinson, Kapil Yadav, Nima Yaghmaei and Ellyn Yakowenko all other respondents of the survey.

## Author Contributions

**Conceptualization:** Tanya Khara, Carmel Dolan.

**Data curation:** Severine Frison, Chloe Angood, Paluku Bahwere, Robert E. Black, André Briend, Nicki Connell, Bridget Fenn, Sheila Isanaka, Philip James, Marko Kerac, Amy Mayberry, Mark Myatt.

**Formal analysis:** Severine Frison, Chloe Angood.

**Funding acquisition:** Tanya Khara, Carmel Dolan.

**Investigation:** Severine Frison, Chloe Angood, Tanya Khara, Paluku Bahwere, Robert E. Black, André Briend, Nicki Connell, Bridget Fenn, Sheila Isanaka, Philip James, Marko Kerac, Amy Mayberry, Mark Myatt, Carmel Dolan.

**Methodology:** Severine Frison, Chloe Angood, Tanya Khara, Paluku Bahwere, Robert E. Black, André Briend, Nicki Connell, Bridget Fenn, Sheila Isanaka, Philip James, Marko Kerac, Amy Mayberry, Mark Myatt, Carmel Dolan.

**Resources:** Carmel Dolan.

**Supervision:** Tanya Khara, Carmel Dolan.

**Validation:** Severine Frison, Chloe Angood, Tanya Khara, Paluku Bahwere, André Briend, Nicki Connell, Bridget Fenn, Sheila Isanaka, Philip James, Marko Kerac, Amy Mayberry, Mark Myatt, Carmel Dolan.

**Visualization:** Severine Frison, Chloe Angood, Robert E. Black.

**Writing – original draft:** Severine Frison.

**Writing – review & editing:** Severine Frison, Chloe Angood, Tanya Khara, Paluku Bahwere, Robert E. Black, André Briend, Nicki Connell, Bridget Fenn, Sheila Isanaka, Philip James, Marko Kerac, Amy Mayberry, Mark Myatt, Carmel Dolan.

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
