## [Decision Letter · Decision Letter 0]

11 Sep 2019

PONE-D-19-20975

Prevention of child wasting: results of a Child Health & Nutrition Research Initiative (CHNRI) prioritization exercise

PLOS ONE

Dear Miss Frison,

Thank you for submitting your manuscript to PLOS ONE. After careful consideration, we feel that it has merit but does not fully meet PLOS ONE’s publication criteria as it currently stands. Therefore, we invite you to submit a revised version of the manuscript that addresses the points raised during the review process (see attached as summary comments with suggestions).

We would appreciate receiving your revised manuscript by Oct 26 2019 11:59PM. To enhance the reproducibility of your results, we recommend that if applicable you deposit your laboratory protocols in protocols.io, where a protocol can be assigned its own identifier (DOI) such that it can be cited independently in the future. For instructions see: http://journals.plos.org/plosone/s/submission-guidelines#loc-laboratory-protocols

We look forward to receiving your revised manuscript.

Kind regards,

Zulfiqar A. Bhutta, PhD

Academic Editor

PLOS ONE

Journal Requirements:

1. In your Methods section, please provide additional information about the participant recruitment method and the demographic details of your participants. Please ensure you have provided sufficient details to replicate the analyses such as: a) the recruitment date range (month and year), and b) a description of any inclusion/exclusion criteria that were applied to participant recruitment ( both for the group which refined the list to 40 items, and for the group who subsequently participated in the prioritisation survey).

Reviewers' comments:

Reviewer's Responses to Questions

**Comments to the Author**

1. Is the manuscript technically sound, and do the data support the conclusions?

Reviewer #1: Yes

Reviewer #2: Partly

2. Has the statistical analysis been performed appropriately and rigorously? 

Reviewer #1: Yes

Reviewer #2: Yes

3. Have the authors made all data underlying the findings in their manuscript fully available?

Reviewer #1: Yes

Reviewer #2: Yes

4. Is the manuscript presented in an intelligible fashion and written in standard English?

Reviewer #1: Yes

Reviewer #2: Yes

5. Review Comments to the Author

Reviewer #1: Very clear and complete research priorization exercise. Be sure to mention the limitation that preference surveys are opinions, and not directives.

Reviewer #2: In addition to comments on the paper uploaded, in the supplementary table, I would recommend removing the 'number' column completely and moving the rank to the left hand side.

6. PLOS authors have the option to publish the peer review history of their article (what does this mean?). If published, this will include your full peer review and any attached files.

Reviewer #1: No

Reviewer #2: Yes: Kerri Wazny

---

## [Author Response · Author response to Decision Letter 0]

22 Oct 2019

Emergency Nutrition Network

 32 Leopold St

Oxford, OX4 1TW

United Kingdom

22nd October 2019

PLoS One

Carlyle House

Carlyle Road

Cambridge, CB4 3DN

United Kingdom

Dear PLoS One Editors,

RE: The prevention of child wasting: Results of a Child Health & Nutrition Research Initiative (CHNRI) prioritisation exercise

Thank you for reviewing our paper. We are grateful to you and the reviewers for your time and inputs. We have addressed the reviewer’s comments and sincerely hope that it will now meet your standards for publication. We appreciate the opportunity to be considered by your journal since we strongly believe that the paper:

 Focuses on prevention of wasting which is of great public health importance. 

 Presents novel results from a Child Health & Nutrition Research Initiative (CHNRI) prioritization exercise that fill a critical gap. 

 Highlights knowledge gaps that have limited our ability to prevent wasting and achieve related global nutrition targets. 

 Provides a five-year investment case for research that could most effectively inform policy change and related on–the-ground practice. 

Please find attached our revised manuscript - and below details of our changes.

Thank you in advance.

Yours sincerely,

Severine Frison (corresponding author on behalf of co-authors)

This paper fills an important knowledge gap on research priorities to prevent wasting in children.

I have some suggestions and comments on the paper, described below:

 I like how the DDDD was put into the context of wasting for the reader – this makes it more accessible. 

Our response: Many thanks for the positive feedback

 The authors describe randomising each research option with the DDDD categories to limit the bias that these categories and the order of appearance of questions have on the scores. Were the DDDD categories themselves randomised? If not, which categories were presented first? What impact might this have had on scoring? 

Our response: The 4Ds were randomised but not the questions within each D. Out of the 18 questions from the description group (the largest D group), the questions ranked highest in terms of RPS and AEA are question 17 and 14 which indicates there was not a bias towards questions presented early on within the category. 

 The AEA formula provided is incorrect. There are 4 questions, not 12 (the formula used is for a 4 criteria exercise, with 3 sub-questions per criteria). The correct formula is below. Additionally, below it should say: 4 is the number of possible scores for each research option across criteria.’

AEA= 1/4 X ∑_(q=1)^4▒□((N(number of scorers with most frequent)/(N(number of scorers)))

Our response: Many thanks for picking that up. Fortunately, this mistake is not reflected in our analysis which uses the correct formula as indicated by the reviewer. We have changed the formula in the text accordingly to match.

 On ethics, while CHNRI did not traditionally require ethics, ethics board are getting stricter and this is not always the case. One better way to justify not going through ethics is to have respondents be co-authors (with group authorship), thereby not having them be participants in the traditional sense, and arguing that it is akin to an organisational strengthening exercise

Our response: Thank you for the suggestion. All participants were asked if they would be happy to be added as group collaborators and the group was added. 

 It would be helpful to state in your summary of participants how quickly they were aborted (e.g. since only 35% of the surveys were completed, were 60% aborted after completing one question? Or were they mostly complete?). You could also define 80% or more as a complete (enough) survey, and say 35% had 100% completeness, but X% had completeness for the purposes of analysis. And exclude any which were very incomplete (from analysis and summary of participants). This may require a more substantial edit, and could change the results depending on the cut-off you use for a ‘complete’ survey.

Our response: Many thanks for this comment. We realise this was unclear. Whilst only 34 out of 146 respondents completed 100% of questions, 113 completed over 80% and 122 completed over 60% of questions. Only a very small number (5) completed 10% or less of the survey questions. Furthermore, over 100 participants responded to each question (from 108 to 131) with slightly more participants for the top 10 questions (from 117 to 131). These numbers have been added to the result section. We hope this brings clarification.

 It’s worth noting that the range of AEA is very high for CHNRI. You discuss this in relation to people belonging to similar groups, etc., but is there any relationship between the rank and AEA and the number of scorers per each RQ and the AEA?

Our response: The higher the AEA in our results, the larger the number of respondents. This probably reflects the fact that the top ranked questions were easier to agree on/”vote for”. More information has been added on number of participants in the result section (see answer to comment 5). There was no relationship between the number of respondents and the rank or between the number of respondents and the level of agreement

 As a preference, I’d move the rank column to the left of the RQ column (so the RQs are numbered on the left in the table)

Our response: This had been changed.

 There appears to be a large number of scorers for the top 10 RQs. Is this because more people scored the highest ranked RQs? Could this be due to the DDDD randomisation (e.g. certain categories being presented first), given only 35% of respondents answered the entire survey? May be useful to discuss this. 

Our response: The Ds were randomised to make sure different categories were presented first for each participant. As mentioned above, there were over 100 participants (min 108 max 131) for each question so we do not feel this had a substantial impact on the ranking.

 I think graphing the AEA is less useful than discussing surprising results – e.g., where there any surprising results within criteria for the top 10? Bottom 10? Any places where RQs seemed to score low across the board? Some that scored high that was a surprise? Anything that’s supported by context?

Our response: Thanks for your comment. We have deleted the figure that did add much and instead highlighted one surprising result: although some discovery questions were very topical, they were ranked low. We discussed top 10 question and the 10 last questions. No other surprising results were observed. 

 There were minor typos throughout the paper. It would be beneficial to have a close read to find and fix these, prior to publication, as I know PLoS does not provide copy-editing. Examples of typos found include:

 Differences in font/font size in some places in the paper

 Incorrect use of ‘;’

 Some spaces missing, particularly before a citation 

 On pg. 11, survey participants (e.g. not surveys) and later, participants should be plural 

 On pg. 18, ‘selection biases can occur’ should be followed by a period not colon

 Some typos in acknowledgements section

Our response: Many thanks for pointing these typos out. We have made corrections accordingly.

---

## [Decision Letter · Decision Letter 1]

9 Jan 2020

The prevention of child wasting: Results of a Child Health & Nutrition Research Initiative (CHNRI) prioritisation exercise

PONE-D-19-20975R1

Dear Dr. Frison,

We are pleased to inform you that your manuscript has been judged scientifically suitable for publication and will be formally accepted for publication once it complies with all outstanding technical requirements.

With kind regards,

Bruno Masquelier, PhD

Academic Editor

PLOS ONE

Additional Editor Comments (optional):

Reviewers' comments:

Reviewer's Responses to Questions

**Comments to the Author**

1. If the authors have adequately addressed your comments raised in a previous round of review and you feel that this manuscript is now acceptable for publication, you may indicate that here to bypass the “Comments to the Author” section, enter your conflict of interest statement in the “Confidential to Editor” section, and submit your "Accept" recommendation.

Reviewer #2: All comments have been addressed

2. Is the manuscript technically sound, and do the data support the conclusions?

Reviewer #2: Yes

3. Has the statistical analysis been performed appropriately and rigorously? 

Reviewer #2: Yes

4. Have the authors made all data underlying the findings in their manuscript fully available?

Reviewer #2: Yes

5. Is the manuscript presented in an intelligible fashion and written in standard English?

Reviewer #2: Yes

6. Review Comments to the Author

Reviewer #2: Thank you for making the edits suggested, I think the paper is great. There are still a few minor typos, if you have the opportunity to do a final proof read.

7. PLOS authors have the option to publish the peer review history of their article (what does this mean?). If published, this will include your full peer review and any attached files.

Reviewer #2: Yes: Kerri Wazny

---

## [Editor Report · Acceptance letter]

5 Feb 2020

PONE-D-19-20975R1 

Prevention of child wasting: results of a Child Health & Nutrition Research Initiative (CHNRI) prioritisation exercise 

Dear Dr. Frison:

I am pleased to inform you that your manuscript has been deemed suitable for publication in PLOS ONE. Congratulations! Your manuscript is now with our production department. 

With kind regards,

on behalf of

Dr. Bruno Masquelier 

Academic Editor

PLOS ONE